# Use of Green Fs Lasers to Generate a Superhydrophobic Behavior in the Surface of Wind Turbine Blades

**DOI:** 10.3390/polym14245554

**Published:** 2022-12-19

**Authors:** Joaquín Rivera-Sahún, Luis Porta-Velilla, Germán F. de la Fuente, Luis A. Angurel

**Affiliations:** Instituto de Nanociencia y Materiales de Aragón, INMA (CSIC-University of Zaragoza), María de Luna 3, 50018 Zaragoza, Spain

**Keywords:** turbine blades, fs lasers, micromachining, superhydrophobicity

## Abstract

Ice generation on the surface of wind generator blades can affect the performance of the generator in several aspects. It can deteriorate sensor performance, reduce efficiency, and cause mechanical failures. One of the alternatives to minimize these effects is to include passive solutions based on the modification of the blade surfaces, and in particular to generate superhydrophobic behavior. Ultra-short laser systems enable improved micromachining of polymer surfaces by reducing the heat affected zone (HAZ) and improving the quality of the final surface topography. In this study, a green fs laser is used to micromachine different patterns on the surface of materials with the same structure that can be found in turbine blades. Convenient optimization of surface topography via fs laser micromachining enables the transformation of an initially hydrophilic surface into a superhydrophobic one. Thus, an initial surface finish with a contact angle ca. 69° is transformed via laser treatment into one with contact angle values above 170°. In addition, it is observed that the performance of the surface is maintained or even improved with time. These results open the possibility of using lasers to control turbine blade surface microstructure while avoiding the use of additional chemical coatings. This can be used as a complementary passive treatment to avoid ice formation in these large structures.

## 1. Introduction

Wind energy is one of the most important renewable energy sources and its implantation in cold climate regions is increasing. Wind generators become more efficient in these regions, because of higher air density [1]. Since humidity is also higher, icing becomes a most relevant problem associated to a significant loss of aerodynamic efficiency. In addition, ice increases both vibration and fatigue loads on the generator blades, giving rise to more extensive maintenance requirements and generating serious safety problems. Several studies have shown that losses associated with ice formation are similar to those related with the remaining wind turbine maintenance issues. It is thus important to minimize ice formation while searching for solutions that facilitate the elimination of ice from the surface of the wind turbine blade [2,3].

Ice protection systems are usually based on two main strategies. These include anti-ice solutions, which prevent ice accumulation on surfaces, as well as de-icing systems, to remove ice from surfaces. The latter are classified into two types of systems: active and passive. Passive methods use the characteristics of the surface, as for instance, generating a hydrophobic surface. This hydrophobic character has been used for many applications, as anti-icing, anticorrosion or self-cleaning properties [4,5,6,7,8]. In the case of wind turbine blades, different coatings have been developed to achieve this property. In many cases, fluorinated compound coated ZnO and/or SiO_2_ nanoparticles are embedded within PDMS substrates [9,10] to enhance a superhydrophobic character to the surface, reducing its free energy. Alternatively, direct addition of polytetrafluoroethylene (PTFE) nanoparticles to these coatings has also been employed for the same purpose [11].

Active anti-icing methods require the input of an external thermal (hot air circulation, electrical resistance heating) or mechanical (surface acoustic waves, moving systems) energy sources [12,13,14] or the use of liquid anti-icing methods. Combinations of these types of solutions have also been reported [15,16]. When considering real applications, most of these passive anti-icing solutions exhibit clear difficulties towards their scale-up. The large size of wind generator blades and the complex ice formation mechanisms, associated to varying atmospheric conditions, are both responsible for the latter difficulty. In view of the former considerations, passive methods are often employed in combination with active ones, in order to reduce the energy required for surface ice removal.

Laser ablation is one of the methods considered to modify the wettability of polymeric materials [4,17,18,19]. Wind turbine blade structures contain a central nucleus fabricated with a glass fiber composite. The latter is protected with gel-coat in order to obtain a smooth, uniform surface finish. Reinforced polyurethanes are considered as one option to fabricate this gel-coat, and laser ablation has been proposed to micromachine and/or nanotexture the surface of polyurethane substrates [19,20,21,22,23,24].

Because of the characteristic thermal properties of polymers, laser irradiation may easily induce their undesired thermal decomposition, thus requiring control of heat accumulation. A similar situation is found when laser treatments are applied to materials with low thermal conductivity and diffusivity, such as ceramics [25] and glass [26,27]. In order to avoid this problem, the use of ultra-short pulse lasers (within the fs pulse regime) working at low pulse repetition frequencies (1–10 kHz) has been proposed.

Usually, when the objective is to obtain superhydrophobic surfaces, laser ablation is combined with the deposition of a specific coating. The laser treatment generates a given micro- and/or nanostructure and the selected coating is deposited in order to modify the chemical properties and enhance the superhydrophobic character of the surface. In fact, the most commonly employed coatings are based on fluorinated compound derivatives [4,17], although the actual environmental restrictions impose the need to develop alternatives. 

The objective of this work is to modify the wettability of a gel-coat surface, frequently used in wind turbine blades, using only a laser micromachining process without employing any additional coating. This could pave the way towards the development of an environmentally friendly technique which would facilitate de-icing in wind generation structures, significantly improving their efficiency.

## 2. Materials and Methods

Experiments were performed on samples that reproduce the structure used in the fabrication of wind turbine blades, fabricated by the company Ingeniería de Compuestos, S.L. (Murcia, Spain). The central part is a 4-mm-thick plate of a fiber glass composite material. The external surface is gel-coat. In the smooth surface, the coating has a thickness of 450 ± 10 μm. As observed in Figure 1, the gel-coat is reinforced with ceramic particles. Some of them are alumina platelets that can reach in-plane dimensions close to 10 μm. The presence of smaller Al and Mg silicate particles is also confirmed.

Laser treatments were performed using a Carbide model (CB3-40W+CBM03-2H-3H, Light Conversion, Vilnius, Lithuania) fs laser in its second harmonic. This linearly polarized green beam output is coupled with a galvanometric mirror configuration controlled by software (Direct Machining Control, Vilnius, Lithuania). This laser system is provided with a pulse on demand mechanism, known as pulse peak divider (PPD). The pulse frequency, *f*, can be adjusted between 1 Hz and 1 MHz, using an appropriate combination of the resonator frequency and the PPD value. During the laser treatments, 249 fs pulses were emitted at 515 nm at a maximum average output power of 12.4 W. The laser beam has a Gaussian beam profile with a diameter 2*r*_0_ = 50 µm, using the 1/e^2^ criterium [28].

The scanning geometry was established as a cross configuration with a set distance between lines, *d*_lines_. The distance between pulses in one line, *d*_pulses_, was fixed by selecting the laser scanning speed, *v*_laser_, for a given value of *f*. Previous reports [29,30] suggested that when *d*_pulses_/*r*_0_ < 0.9, the energy along the beam scanned line could be considered constant and takes the following value at its center:(1)$$F_{\mathrm{center}}=1.588\frac{\pi \;r_{0}}{2\;d_{\mathrm{pulses}}}\;F_{\mathrm{pulse}},$$
where *F*_pulse_ is the average fluence associated with a single pulse. The total energy deposited in a given area during scanning along a line, $❬F_{1D}❭$, can be calculated as:(2)$$❬F_{1D}❭=\frac{\pi \;r_{0}}{2\;d_{\mathrm{pulses}}}\;F_{\mathrm{pulse}}=N_{1D}\;F_{\mathrm{pulse}}.$$
*N*_1D_, the ratio between $❬F_{1D}❭$ and *F*_pulse_, can be considered as the effective number of pulses in a given position.

Surface topography was characterized using confocal microscopy (2300 Plμ Sensofar, Terrassa, Spain). Surface morphology was also analyzed with a field-emission scanning electron microscope (FESEM, MERLIN Carl Zeiss GmbH, Oberkochen, Germany) using secondary electron (SE), in-lens, and backscattered (ESB) detectors. The electron beam acceleration voltage was set to 5 kV. Chemical surface characterization was performed using energy dispersive X-ray spectroscopy (EDS, INCA350 Oxford Instruments).

Contact angle measurements were performed in a home-made system using a micropipette (model Multipette^®^ E3x, Eppendorf AG, Hamburg, Germany), coupled with a camera (model UI-3080CP Rev.2, IDS, Obersulm, Germany) and a teleobjective (Thorlabs, Germany). 6 μL droplets were deposited on the surfaces and their shape was recorded after 1 min. Contact angle values were obtained from the photographs using the free software ImageJ with installed LBADSA (Low Bond Axisymmetric Drop Shape Analysis) plugin [31].

Raman spectroscopy was also used for structural investigation before and after laser machining. The apparatus used in this work was a Jasco NRS 3100 equipped with two lasers at 785 and 532 nm, two diffraction gratings (600 and 1800 gr/mm), and three objectives (×5, ×20 and ×100) and a motorized stage with step accuracy of 1 μm with spatial resolution of 8 μm^3^ and spectra resolution of 1 cm^−1^. Samples were measured with the following instrument set up: Laser 532 nm, grating 600 gr/mm, and objective X100.

## 3. Results and Discussion

### 3.1. Definition of the Laser Ablation Parameters

Initial laser treatments were carried out in order to define the main ablation parameters that characterize the laser interaction with this material. A set of lines were patterned with *d*_lines_ = 200 μm in an area of 6 × 6 mm^2^. This *d*_lines_ value was selected in order to avoid a significant thermal interaction between two consecutive lines during the laser treatment. Thermal incubation was also minimized [27] by working with low frequency values, in this case 10 kHz, and *v*_laser_ = 150 mm/s, yielding *d*_pulses_ = 15 μm. The criterium of working with a uniform Gaussian energy distribution along the line is fulfilled by applying these processing parameters, where *d*_pulses_/*r*_0_ = 0.6 < 0.9. The selected energy per pulse was set at *E*_pulse_ = 48.5 μJ/pulse, reaching a total average fluence value of $❬F_{1D}❭$ = 6.5 J/cm^2^.

Figure 2a shows the topography of the surface. The width of the machined region at the sample surface is approximately 27 μm in the initially processed horizontal lines. For the vertical lines, the width increases up to 32 μm. In both line orientations, the depth of the machined volume is low, approximately 3 μm. This increase in the ablated region size during the second scan was observed in all the treatments and it is due to the change in absorption of the sample surface during the two scans, a phenomenon described as incubation. The horizontal scan was performed over the original surface. The vertical scan was performed over a surface with the roughness generated by the previous laser scan, increasing the energy absorption and making the laser treatment more effective. 

After measuring the contact angle values, it was observed (Figure 2b) that the hydrophobicity of the surface improves and the contact angle increases from values ca. 68° to about 97°.

Similar microstructures were generated reducing the distance between lines from 200 to 35 μm. Line to line overlap begins with this latter condition. As observed in Figure 2b, the contact angle increases while the distance between machined lines is reduced, reaching a value of 119° for *d*_lines_ = 35 μm. The insets show the shape of the water droplet in the contact angle measurements. More detailed images can be observed in the Appendix A.

### 3.2. Changes in Surface Color

Figure 3 shows the aspect of the sample surface after these laser treatments. The surface now appears darkened, as reported in previous works for the achievement of laser-induced marking effects [22,32]. Usually, the marking treatments were performed using a near infrared laser and a laser sensitive additive, such as BiOCl [22] or Bi_2_O_3_ [32]. In these cases, the laser process induces formation of laser-heated micron size areas where polyurethane is decomposed by carbonization associated to the pyrolysis of the polymer chains.

In order to go into more detail about the origin of this phenomenology, additional experiments were performed increasing the level of $❬F_{1D}❭$. Figure 4 shows the aspect of the bottom of a machined region after processing with *E*_pulse_ = 48.5 μJ/pulse and *v*_laser_ = 9 mm/s, conditions that lead to $❬F_{1D}❭$ = 107.7 J/cm^2^, 16.7 times higher than those used in the processing of the sample presented in Figure 2 and Figure 3. Observation with an in-lens detector, as shown in Figure 4a, suggests that irradiation with a fs green laser induces very limited thermal effects on the polymeric material. Spherical nanoparticles are not associated with molten material. The white contrast on the EBS image of the same region (Figure 4b) suggests that the latter are part of the ceramic reinforcement of the material.

Additional experiments were repeated performing a laser treatment in an Ar atmosphere instead of air. A similar change in color was observed (see Appendix A), indicating that surface darkening is not generated by oxidation of hydrocarbons of the polymer chains, although these could take place during the laser treatment in air. Raman spectra were recorded at the bottom of the machined grooves in both laser-treated regions and the results are compared with the original surface in Figure 5. In all the cases, the band associated with Al_2_O_3_ at 631 cm^−1^ is clearly observed. Polymer bands in the laser treated regions are broader, suggesting partial decomposition of the polymer, but the typical broad diffusion band in the 1000–2000 cm^−1^ range, usually assigned to amorphous carbon, is not so evident. This is also an indication that laser irradiation has induced only a limited thermal accumulation effect. Spectra in both treatments show similar trends, indicating that the atmosphere under which laser irradiation was performed is not determinant. The green fs laser irradiation results in a direct photo-induced chemical decomposition of the polymer.

### 3.3. Optimising Contact Angle Values

From the analysis of the previous results, it was decided to work directly on air but modifying the laser conditions, in order to increase the volume of material that is eliminated during the laser treatment. The approach followed a reduction of the laser beam scanning speed. This was combined with an optimization of the distance between machined lines.

Figure 6 shows the evolution of the contact angle for different laser scanning speeds. A reduction of *v*_laser_, reduces *d*_pulses_ and increases $❬F_{1D}❭$. With the laser parameters used in this study, $❬F_{1D}❭$ ranges from $❬F_{1D}❭\;$= 6.5 J/cm^2^ with *v*_laser_ = 150 mm/s to $❬F_{1D}❭$ = 194.0 J/cm^2^ with *v*_laser_ = 5 mm/s. These data show a condition in which superhydrophobic contact angle values (163°) have been reached: *v*_laser_ = 5 mm/s and *d*_lines_ = 75 μm. Another interesting aspect is that this surface also exhibits a lower rolling angle, observed below 20°.

The topography of the sample that exhibits this superhydrophobic behavior is presented in Figure 7a. The same phenomena that were explained for the topography of the sample presented in Figure 2 are also observed in this case. Machining is more effective in the last vertical scan. Figure 7b shows a vertical linear profile that measures the depth of the grooves machined in the horizontal direction. These laser machining conditions produce grooves that reach a depth of approximately 120 ± 10 μm. In the horizontal direction profiles, vertical grooves with depths up to 180 μm were measured.

Several processing parameters were explored around these conditions, which produced a surface with contact angles above 150°. Figure 8 shows a set of 21 experimental conditions that also exhibit this behavior, using laser pulses with *E*_pulse_ = 48.5 μJ/pulse in all cases. Contact angles above 150° were achieved when the distance between lines was in the range between 50 and 80 μm. Smaller *d*_lines_ values, in the range between 25 and 50 μm, were also explored. In these cases, it was observed that consecutive machined lines start to overlap, and the measured contact angles decrease to values below 150°.

Three additional series of samples were processed in order to analyze in more detail the geometric parameters that facilitate obtaining surfaces with high contact angle values. Initial series correspond to a set of surface microstructures performed decreasing *d*_lines_ from 200 μm to 40 μm with the rest of laser parameters fixed: *E*_pulse_ = 48.5 μJ/pulse, *v*_laser_ = 9 mm/s and $❬F_{1D}❭$ = 108 J/cm^2^. As shown in Figure 9, the generated square lattice is very well defined with machined lines that have a uniform width of 45 ± 4 μm. The width of these lines is similar in all the samples until *d*_lines_ = 50 μm, geometry where the machined regions start to overlap in some positions. This overlap starts earlier than expected due to the irregularities that the alumina particles generate in the machining process, as indicated later. The depth reaches a value of 44 ± 5 μm in the horizontal lines, and it increases to 63 ± 4 μm in the vertical ones. When the machined lines overlap, *d*_lines_ < 50 μm (Figure 9d), the difference between the two orientations leads to a geometry with a high anisotropy.

Figure 10a shows the evolution of the contact angle values in this series of samples. It increases while *d*_lines_ is reduced, until the grooves start to overlap. This phenomenology suggests that a superhydrophobic surface can be achieved in this material by controlling the size of the pillars of the original surface that remains between the machined grooves, together with their depths. The inset presented in Figure 10a shows the profile measured in the sample processed with *d*_lines_ = 65 μm. It can be approximated with a trapezoidal profile in both directions. A scheme of the different profiles can be observed in Figure 10b. Before overlapping, the profiles have the trapezoidal profile, maintaining the size of *b* and *h* and modifying the size of the pillar, *c*. When the machined lines overlap (case of *d*_lines_ = 40 μm), the profile exhibits a triangular geometry, reducing the depth of the resulting machined grooves.

Two additional series were processed modifying the energy per pulse or the laser scanning speed. In the first case, due to the gaussian energy distribution in the laser beam, *b* and *h* increase when *E*_pulse_ increases, maintaining *c* + *b* = 65 μm. In the second case, *E*_pulse_ is constant, and it is expected that *b* and *c* will be similar in all cases, increasing the depth of the machined region when the laser scanning speed is reduced. 

Figure 11 shows the surface microstructure changes when different regions of the sample were processed with *v*_laser_ = 9 mm/s and *d*_lines_ = 65 μm and the energy per pulse was modified from 13.6 μJ/pulse ($❬F_{1D}❭$ = 30.1 J/cm^2^) to 48.4 J/pulse ($❬F_{1D}❭$ = 107.7 J/cm^2^). An additional sample was processed with *E*_pulse_ = 8.8 μJ/pulse, $❬F_{1D}❭$ = 19.4 J/cm^2^, but this level of energy was too low to obtain uniform machining along the lines, as observed in Appendix A. Table 1 collects the evolution in the width and depth of the machined regions that can be derived from these FESEM micrographs. It reflects the gaussian energy distribution in the direction perpendicular to the laser scan [29,30]. The values for *c* and *b* are similar in the horizontal and vertical directions in all cases. By contrast, differences are observed in the depth of the machined regions. For low *E*_pulse_ values, *h*_h_/*h*_v_ = 1 and this ratio increases up to 1.4 for the high *E*_pulse_ values. 

Debris are generated during the laser process and are deposited on the sample surface. In order to explore if these particles modify the surface wetting properties, additional samples were prepared, and they were cleaned in an ultrasonic bath during a 10 s period. The latter treatment eliminates the debris, and upon drying it was observed that contact angles were similar both before and after cleaning. 

It was also observed that the depth of the machined region is nearly proportional to ❬*F*_1D_❭. Figure 12a shows this dependence in the case of *h*_v_. Combining these measurements with the contact angle values, it can be deduced that a minimum depth of approximately 22 μm is required to reach a contact angle above 150° (Figure 12b).

Finally, FESEM micrographs show that when low *E*_pulse_ values are used and the groove depth is low, laser irradiation only machines the polymeric material without affecting the ceramic particles that are clearly observed in the micrographs, generating additional inhomogeneities in the borders of the non-machined regions.

A new series of samples was fabricated using *E*_pulse_ = 48.5 μJ/pulse and *d*_lines_ = 65 μm and increasing the laser scanning speed from 9 mm/s to 45 mm/s. In this case, pulse overlap within a line was decreased, reducing the ❬*F*_1D_❭ values. In this series, it was expected that *c* and *b* would be similar in all cases and that only the depth of the machined grooves would evolve, modifying the roughness of the surface. FESEM micrographs of the samples processed with the new *v*_laser_ values are presented in Figure 13. The micrograph of the sample processed with *v*_laser_ = 9 mm/s is shown in Figure 11f. The values of *b* and *c* are similar in all cases and correspond to the values of the sample processed with *E*_pulse_ = 48.5 μJ/pulse, included in Table 1. Further comparison of the evolution of *h*_v_ and of the contact angle is presented in Figure 12. The evolution of *h*_v_ suggests that this magnitude is determined mainly by ❬*F*_1D_❭ and that the minimum depth required to obtain contact angles above 150° is approximately 22 μm. It is important to define this value in order to minimize the changes in the surface roughness, as well as their potential influence on the aerodynamic performance of the wind turbine [33]. The average roughness value measured on these surfaces with *h*_v_
≈ 22 μm is below 10 μm.

The effect of surface structure on wetting properties has been extensively studied [34,35], following the ideas proposed by Wenzel [36] and by Cassie and Baxter [37]. Usually, when a water droplet is deposited on a rough surface, two wetting states can take place. In the noncomposite state, the liquid penetrates into the troughs of the rough surface and the contact angle value, $\theta_{\mathrm{rW}}$, can be described by the Wenzel equation [36]:
(3)$$cos\theta_{\mathrm{rW}}=r\;cos\theta$$
where *θ* is the contact angle determined on the flat surface and *r* is the roughness factor, defined as the ratio of the surface area to the geometrically projected one. In the case of the geometry presented in Figure 10b, it can be calculated as:(4)r=((dlines+ch)hh2+(bh2)2)+((dlines+cv)hv2+(bv2)2)+chcvdlines2

In the second state, called composite state, air is entrapped in the troughs of the rough surface. The contact angle value, $\theta_{\mathrm{rC}}$, can be calculated using the Cassie–Baxter model [37,38]. The liquid does not entirely wet the micromachined surface because air is trapped in the grooves generated during the laser process. Following the ideas proposed by this model, larger contact angles are reached if the interface area between the solid and the liquid is low, increasing the region where water is in contact with air.
(5)$$cos\theta_{\mathrm{rC}}=A\left(cos\theta +1\right)-1$$
where *A* is defined as the solid–liquid contact area fraction of the substrate [17].
(6)$$A=\frac{c_{h}\;c_{v}}{d_{\mathrm{lines}}{}^{2}}\;$$

Table 2 shows the values of *A* and *r* calculated using Equations (4) and (6) and the geometrical parameters defined in each of the microtextured surfaces of the two series. In the case of the first series, the modification of the machined region width is reflected in the evolution of *A*, and the changes in *h* are reflected in *r*. In the case of the second series, *A* is the same for all the samples, and the change in *h* modifies the value of the roughness factor. Figure 14 shows the evolution of the contact angle cosine as a function of the estimated values of *r* and *A*. In the case of *r* (Figure 14a), two different regions are observed. One corresponding to $cos\theta_{\mathrm{rW}}$ < −0.8, where $\theta_{\mathrm{rW}}$ < 143° and *h* < 19 μm (Figure 12), is described more appropriately by the Wenzel model. In the second series, *A* is the same for the five different conditions (blue points in Figure 14b, *A* = 0.10). By contrast, the contact angle evolves, being similar only in the two surfaces processed with *v*_laser_ = 9 mm/s and 18 mm/s. This result suggests that the contact angle depends on the surface roughness for low *h* values and this dependence is lost when this depth increases. In consequence, when $cos\theta_{\mathrm{rC}}$ > −0.8, where $\theta_{\mathrm{rW}}$ > 143° and *h* > 19 μm, the behavior approaches the Cassie–Baxter model, showing the influence of A in the behavior of the sample (Figure 14b).

In consequence, we can estimate that the surface roughness plays an important role when the depth of the machined grooves is below 19 μm. Obviously, the laser treatment strongly modifies the surface characteristics. The thermodynamic properties of the machined grooves are completely different from those areas that have not been modified by irradiation. This limits the applicability of these models. These limitations have also been reported in several works [39,40]. However, the conclusion that a minimum depth is required to entrap air in the troughs of the rough surface is valid. So is the fact that the latter conditions are needed in order to achieve superhydrophobic behavior.

### 3.4. Time Evolution of Contact Angle Values

When similar laser treatments were performed in metallic surfaces, it was observed that wetting properties evolved with time due to the chemical absorption of organic compounds form air moisture [6,41]. Time evolution was also analyzed in order to determine if a similar process is followed by these organic surfaces. The contact angle in 15 surfaces that exhibited a superhydrophobic behavior was measured the day after the samples were processed, one month later and after six months. During these time periods, samples were covered with lens cleaning tissues and placed into a plastic PE bag. The bag was stored in the laboratory at room temperature, in the period between the months of July and November. Table 3 shows the measured contact angles for all these samples. The initial ten samples correspond the conditions arranged in the two horizontal lines presented in Figure 8. The last five surfaces were processed with *d*_lines_ = 72 μm, *v*_laser_ = 9 mm/s and reducing the energy per pulse of the laser from 48.5 μJ/pulse to 24.3 μJ/pulse. Most of them exhibit an increase in contact angle values after the first month. This improvement is maintained or even increased when the static contact angles are measured after six months. It is important to mention the two cases where a contact angle was measured above 175°, because these two surfaces exhibit rolling angles lower than 10°.

## 4. Conclusions

Fs green laser systems can be used to machine composite materials that are covered with a gel coat layer, usually employed in wind generator blades. After an adequate selection of laser processing parameters aiming to minimize heat accumulation, it is possible to obtain a well-defined micropattern on the surface of the material with minimal thermal effects in the regions where the laser beam has not irradiated the surface.

Combining an adequate selection of the interlinear distance, *d*_lines_, and the laser scanning speed, *v*_laser_, it is possible to modify the surface wettability. Static contact angle values above 150° may be achieved by applying processing parameters which result in an adequate selection of the generated micropillars and the depth of the machined grooves.

During the laser treatment, a change in the surface color is generated even when laser irradiation is carried out in an Ar atmosphere. This suggests that the laser-triggered decomposition of the polymer is not associated to thermal effects, nor to oxidation.

The analysis of the results suggests that the minimum depth of the machined grooves required to reach static contact angles above 150° is approximately 22 μm. It is important to work with this minimum depth in order to minimize surface roughness modification.

It was also been observed that contact angle values evolve with time, increasing in some cases by more than 10° after one month. Further evolution has also been measured during a period of six months. In some cases, the obtained surfaces reach rolling angles lower than 10°. This phenomenon requires further investigation, in order to determine the evolution of the surface chemical properties as a function of time.

The surface properties herein reported have been achieved without any additional modification of the surface. They pave the way to include superhydrophobic surface character engineering in anti-icing strategies which may contribute to avoid wind generator performance degradation.

## Figures and Tables

**Figure 1 polymers-14-05554-f001:**
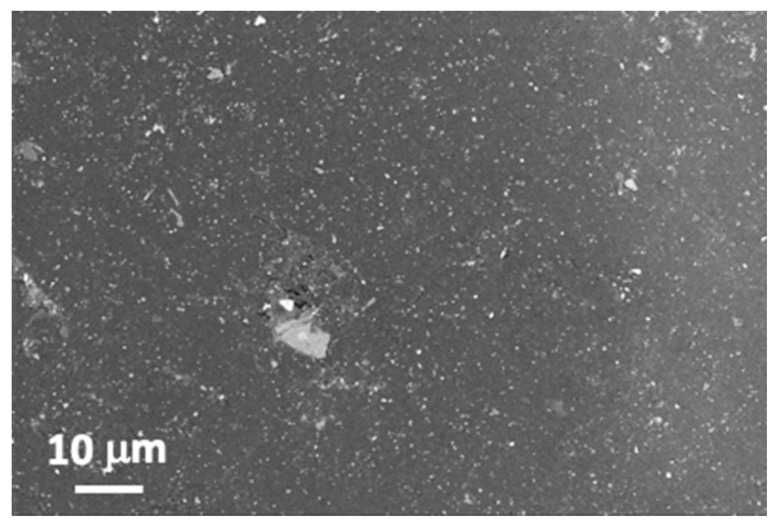
FESEM image with EBS detector of the original sample surface employed in this work. Gray particles correspond to ceramic phases.

**Figure 2 polymers-14-05554-f002:**
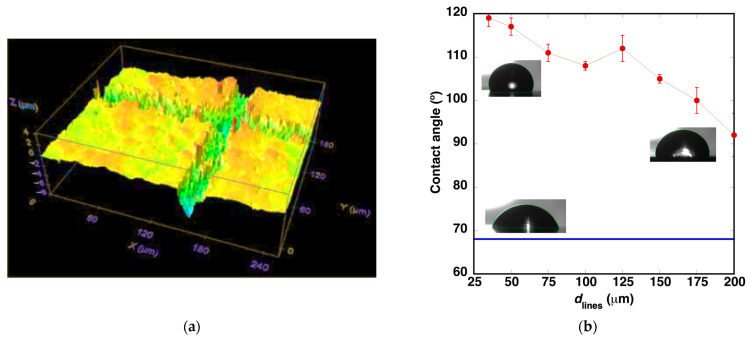
(**a**) Topography of the surface after laser machining for *d*_lines_ = 200 μm. (**b**) Evolution of the contact angle values with the reduction of *d*_lines_. The horizontal blue line indicates the value of 68° measured in the original surface, before laser microachining. Insets show the images obtained for measuring the contact angle in the original surface, and with *d*_lines_ = 200 μm and *d*_lines_ = 35 μm.

**Figure 3 polymers-14-05554-f003:**
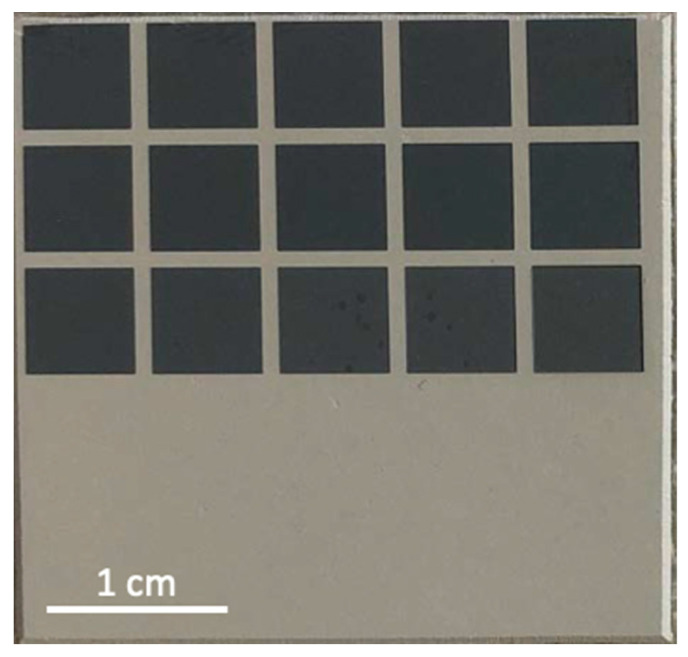
Photograph of the sample surface subjected to study, where laser treatments were carried out in air.

**Figure 4 polymers-14-05554-f004:**
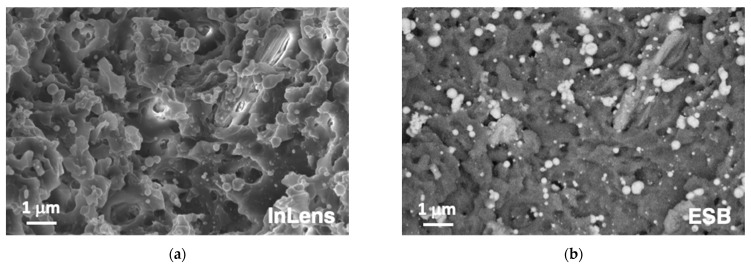
FESEM images recorded with In Lens (**a**) and EBS (**b**) detectors at the bottom of a laser machined groove processed with *E*_pulse_ = 48.5 μJ/pulse, *v*_laser_ = 9 mm/s and *d*_lines_ = 60 μm.

**Figure 5 polymers-14-05554-f005:**
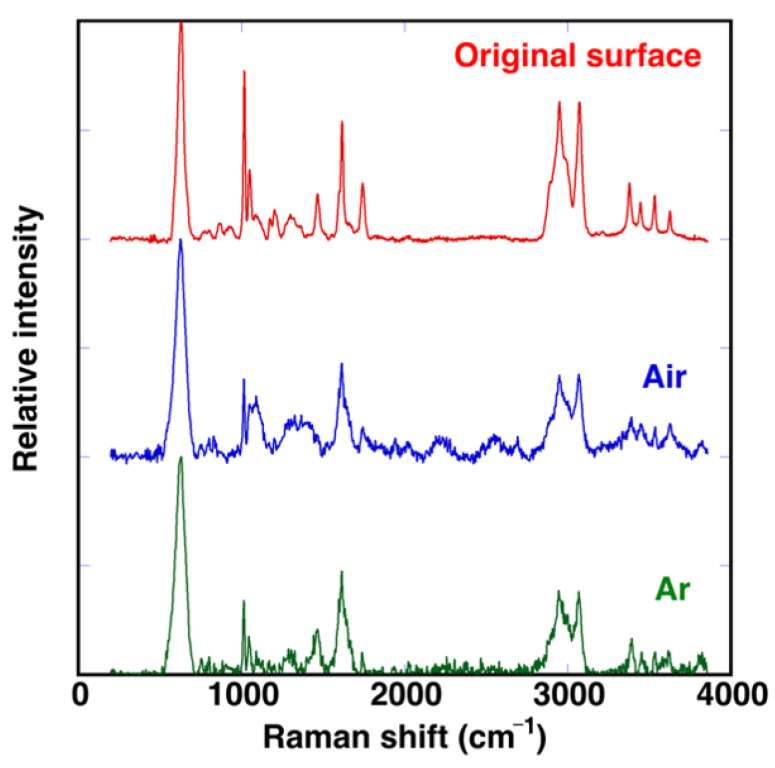
Raman spectra recorded on the original surface and on the bottom of a laser machined groove on samples processed in air and in Ar with *E*_pulse_ = 48.5 μJ/pulse, *v*_laser_ = 9 mm/s and *d*_lines_ = 65 μm.

**Figure 6 polymers-14-05554-f006:**
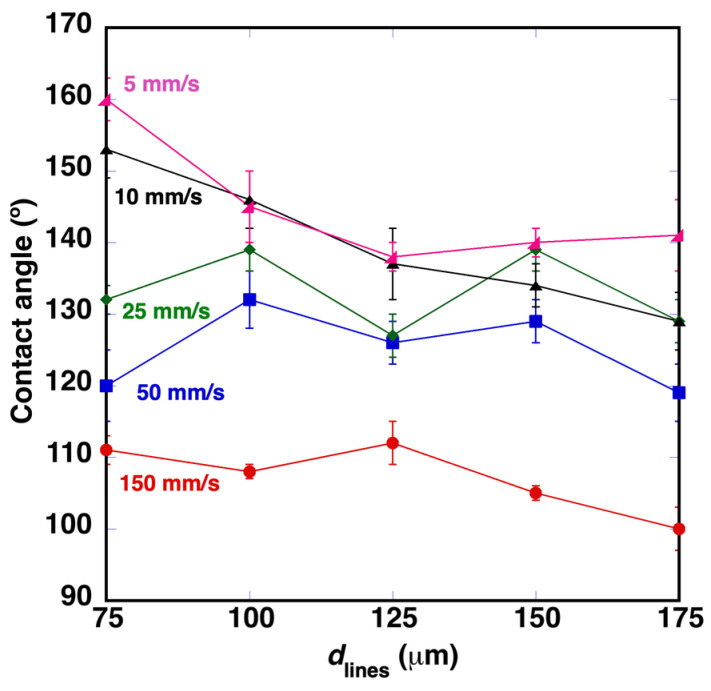
Evolution of contact angle values for different distances between machined lines and different laser scanning speeds.

**Figure 7 polymers-14-05554-f007:**
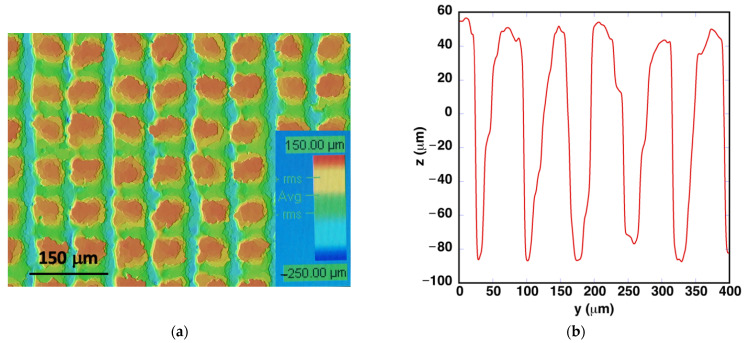
(**a**) Surface topography and (**b**) line profile in the vertical direction in the sample processed with *E*_pulse_ = 48.5 μJ/pulse, *v*_laser_ = 5 mm/s and *d*_lines_ = 75 μm.

**Figure 8 polymers-14-05554-f008:**
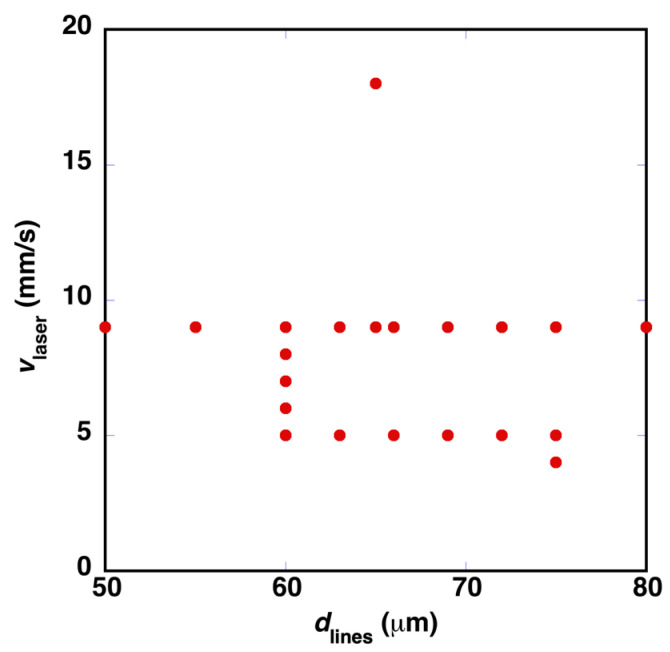
Different combinations of *d*_lines_ and *v*_laser_ (*E*_pulse_ = 48.5 μJ/pulse) that have generated a surface with contact angles above 150°.

**Figure 9 polymers-14-05554-f009:**
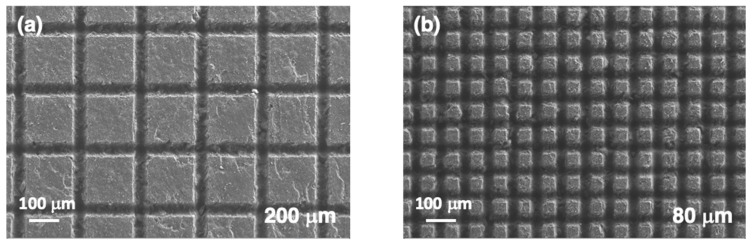
FESEM images of the surface of the samples with different *d*_lines_ values: (**a**) 200 μm, (**b**) 80 μm, (**c**) 55 μm and (**d**) 40 μm. The rest of the laser processing parameters were fixed: *E*_pulse_ = 48.5 μJ/pulse, *v*_laser_ = 9 mm/s and $❬F_{1D}❭$ = 108 J/cm^2^.

**Figure 10 polymers-14-05554-f010:**
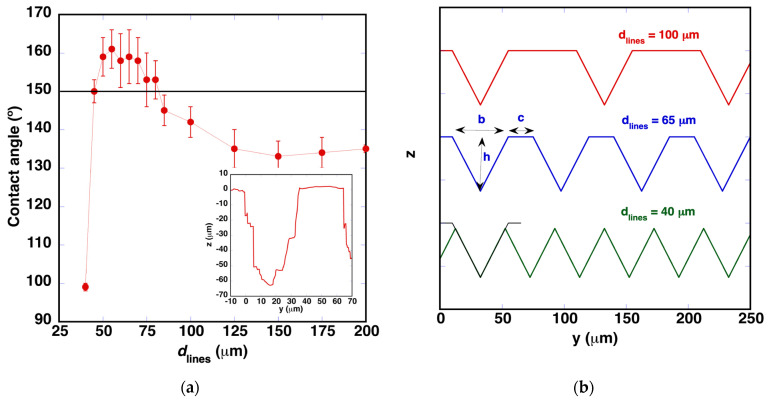
(**a**) Evolution of the contact angle values with the reduction of *d*_lines_. The horizontal line indicates 150°. The rest of the laser processing parameters were fixed as follows: *E*_pulse_ = 48.5 μJ/pulse, *v*_laser_ = 9 mm/s and $❬F_{1D}❭$ = 108 J/cm^2^. The inset shows the profile observed in the sample processed with *d*_lines_ = 65 μm. (**b**) Scheme of the different types of profiles observed for two cases where grooves do not overlap (*d*_lines_ = 100 μm, *d*_lines_ = 65 μm) and for a last one (*d*_lines_ = 40 μm), where they overlap. Black lines show one of the previous profiles in order to show the reduction in *h*.

**Figure 11 polymers-14-05554-f011:**
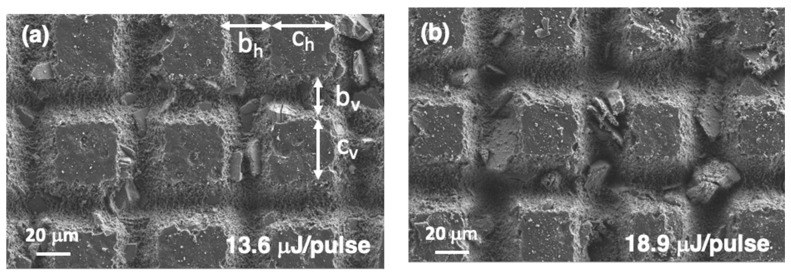
FESEM images of the surface of the samples with different *E*_pulse_ values: (**a**) 13.6 μJ/pulse, (**b**) 18.9 μJ/pulse, (**c**) 24.3 μJ/pulse, (**d**) 31.0 μJ/pulse, (**e**) 38.7 μJ/pulse, and (**f**) 48.5 μJ/pulse. The rest of the laser processing parameters were fixed as follows: *v*_laser_ = 9 mm/s and *d*_lines_ = 65 μm.

**Figure 12 polymers-14-05554-f012:**
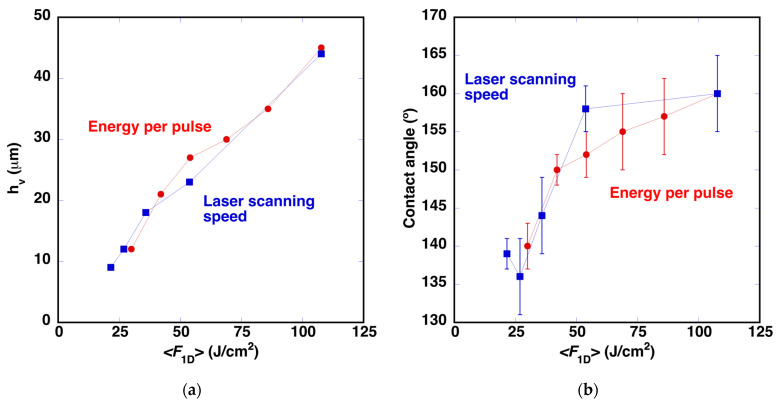
(**a**) Evolution of *h*_v_ as a function of ❬*F*_1D_❭, in a series of experiments where the energy per pulse has been modified. The plot includes a second sample series fabricated with different laser scanning speeds. (**b**) Evolution of the contact angle dependence for both series of samples.

**Figure 13 polymers-14-05554-f013:**
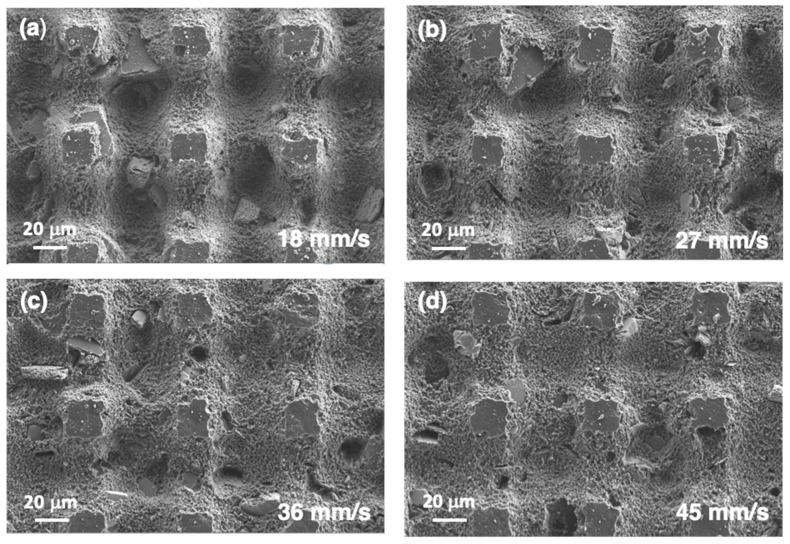
FESEM micrographs obtained on the sample’s surfaces treated with different *v*_laser_ values: (**a**) 18 mm/s, (**b**) 27 mm/s, (**c**) 36 mm/s, and (**d**) 45 mm/s. The rest of the laser processing parameters are the same in all cases: *E*_pulse_ = 48.5 μJ/pulse and *d*_lines_ = 65 μm.

**Figure 14 polymers-14-05554-f014:**
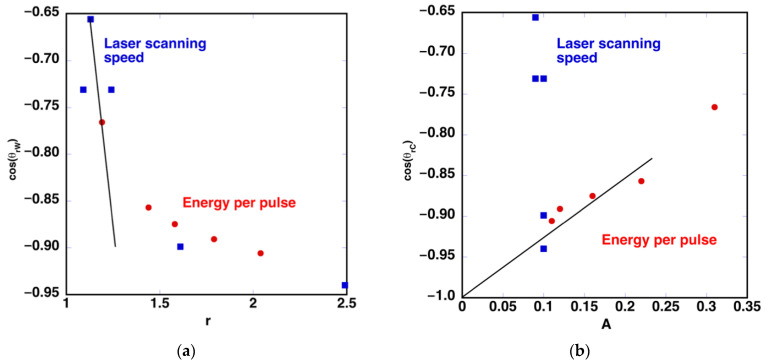
(**a**) Relation between $cos\theta_{\mathrm{rW}}$ and *r*. (**b**) Relation between $cos\theta_{\mathrm{rC}}$ and *A*. Lines are eye-guides.

**Table 1 polymers-14-05554-t001:** Differences in the microstructures generated with treatments using different *E*_pulse_ values and maintaining *v*_laser_ = 9 mm/s and *d*_lines_ = 65 μm. Definitions for *b*_h_, *c*_h_, *b*_v_ and *c*_v_ are presented in Figure 11a. Values for *h*_h_ and *h*_v_ correspond to machined groove depths.

*E*_pulse_ (μJ/pulse)	*b*_h_ (μm)	*c*_h_ (μm)	*h*_h_ (μm)	*b*_v_ (μm)	*c*_v_ (μm)	*h*_v_ (μm)	Contact Angle
48.5	45 ± 1	20 ± 1	63 ± 3	45 ± 1	21 ± 1	44 ± 2	160° ± 5°
38.7	44 ± 1	22 ± 1	48 ± 2	44 ± 1	22 ± 1	34 ± 3	155° ± 5°
31.0	41 ± 1	24 ± 1	38 ± 1	42 ± 1	22 ± 1	30 ± 2	153° ± 5°
24.3	39 ± 1	26 ± 1	26 ± 2	39 ± 2	26 ± 1	27 ± 2	151° ± 3°
18.9	34 ± 1	30 ± 1	21 ± 1	34 ± 1	31 ± 1	21 ± 1	149° ± 2°
13.6	28 ± 2	35 ± 2	12 ± 1	28 ± 2	37 ± 1	12 ± 1	140° ± 3°

**Table 2 polymers-14-05554-t002:** Values of *A* (Cassie-Baxter model) and *r* (Wenzel model) in the two series of laser processed samples modifying *E*_pulse_ or *v*_laser_.

** *E* _pulse_ ** **(μJ/pulse)**	13.6	18.9	24.3	31.0	38.7	48.5
*A*	0.31	0.22	0.16	0.12	0.11	0.10
*r*	1.19	1.44	1.58	1.79	2.04	2.49
** *v* _laser_ ** **(mm/s)**		45	36	27	18	9
*A*		0.10	0.10	0.10	0.10	0.10
*r*		1.09	1.13	1.24	1.61	2.49

**Table 3 polymers-14-05554-t003:** Time evolution for static contact angle values measured in 15 different laser treated surface areas characterized as superhydrophobic.

*E*_pulse_ (μJ/pulse)	*v*_laser_ (mm/s)	*d*_lines_ (μm)	Contact Angle 1 Day	Contact Angle 1 Month	Contact Angle 6 Months
48.5	5	75	159°	167°	167°
48.5	5	72	161°	172°	168°
48.5	5	69	160°	173°	168°
48.5	5	66	161°	167°	170°
48.5	5	63	159°	157°	164°
48.5	9	75	158°	167°	170°
48.5	9	72	159°	159°	175°
48.5	9	69	158°	173°	172°
48.5	9	66	158°	165°	>175°
48.5	9	63	158°	169°	>175°
43.5	9	72	157°	171°	171°
38.6	9	72	156°	167°	158°
35.7	9	72	153°	166°	159°
31.0	9	72	151°	157°	159°
24.3	9	72	152°	157°	160°

## Data Availability

Additional data are provide in the Appendix A.

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
