# Peer review of "Use of Green Fs Lasers to Generate a Superhydrophobic Behavior in the Surface of Wind Turbine Blades"

_polymers, 2022, doi:10.3390/polym14245554_

Round 1

Reviewer 1 Report

The paper is very original and very interesting.

The authors adopted the appropriate research methodology.

 Analysis and interpretation of the obtained research results at a very good substantive level.

The paper was written correctly.

The language of the paper is satisfactory.

The authors drew the correct conclusions from the obtained research results.
The paper meets the journal's requirements and can be published.

Reviewer 2 Report

In this study, the surface was ablated using a femtosecond laser and the change in hydrophobicity was measured.

1. The shape (width, depth) of the ablated pattern is a very important factor in hydrophobicity. In particular, these parameters should be considered for application to hydrophobic models. Data on shape change according to each processing condition can be added. 

2. As debris is generated during the ablation process deposited debris on the surface can be observed in the image, which affects the hydrophobicity. Have you performed any experiments on this?

3. What is the purpose of Section 3.3? Does the chemical properties of a surface change over time?

4. Would you explain why the hydrophobicity had increased after 1 month? (from 1 day to 1 month)

5. What are some other ways to increase the hydrophobicity of turbine blades? And what is the difference between the hydrophobicity of these coated surfaces and the processed (ablated) surfaces?

6. Does the ablation process affect the efficiency of the turbine blade?

7. Due to the nature of laser processing, the topography of the pattern has a slope (not steep), and this affects hydrophobicity. Did you consider this when applying it to the hydrophobicity model?

Reviewer 3 Report

This manuscript has great innovative significance in investigating green fs lasers to generate a superhydrophobic behavior in the surface of wind turbine blades. The work can arouse wide interests of researchers in design and preparation of new functional materials. The manuscript is interesting. In my frank opinion, the manuscript should be deserved for its final publication in such high-level Journal. The main reasons are as follows:

1. At first, the English ABSTRACT should be revised, and a unified simple present tense should be used.

2. The research significance and future work should be described in the final stage of the abstract.

3. Aims need to be concisely stated and added at the end of introduction. Not only what was done/investigated, but why.

4. In Fig.2b, If possible, please provide error bars.

5. In Fig.6, If possible, please provide error bars.

6. Under normal conditions, in conclusion section, important conclusions should be elaborated point by point for brevity and prominence, such as a) … … b) … … c) … ….

7. And also in the last point future research work should be given in conclusion section.

8. The reference is a little outdated, please update it. As seen in introduction about “e, generate a hydrophobic surface [4-5]”, such as:

A)- Tian Shi, Jingsong Liang, Xuewu Li, Chuanwei Zhang, Hejie Yang. Improving the Corrosion Resistance of Aluminum Alloy by Creating a Superhydrophobic Surface Structure through a Two-Step Process of Etching Followed by Polymer Modification. Polymers 2022, 14(21), 4509.

In such reference, the above reference should be quoted and added for great correlation with marine ecosystem corrosion.

Round 2

Reviewer 2 Report

Thank you for your response.